# Superfine Marigold Powder Improves the Quality of Sponge Cake: Lutein Fortification, Texture, and Sensory Properties

**DOI:** 10.3390/foods12030508

**Published:** 2023-01-22

**Authors:** Si-Yeon Kim, Seok-Young Hong, Hyun-Su Choi, Jong-Hun Kim, Se-Ho Jeong, Su-Yong Lee, Sung-Hun Kim, Dong-Un Lee

**Affiliations:** 1Department of Food Science and Technology, Chung-Ang University, Anseong 17546, Republic of Korea; 2Carbohydrate Bioproduct Research Center, Department of Food Science & Biotechnology, Sejong University, Seoul 05006, Republic of Korea; 3Smart Farm R&D, WOOREE Green Science Co., Ltd., Ansan 15409, Republic of Korea

**Keywords:** jet mill, lutein-enriched marigold powder, baking, texture profiles, sponge cake, sensory evaluation

## Abstract

This study aimed to investigate and optimize the quality and sensory properties of baked products with lutein-enriched marigold flower powder (MP). Lutein-enriched marigold flowers produced via hydroponic methods using LED lights were used as a functional material in sponge cakes to increase lutein content. MP particles were divided into coarse (Dv_50_ = 315 μm), fine (Dv_50_ = 119 μm), and superfine MP (Dv_50_ = 10 μm) fractions and added to the sponge cake after being designated to control (sponge cake prepared without MP), coarse MPS (sponge cake prepared with coarse MP), fine MPS (sponge cake prepared with fine MP), and superfine MPS (sponge cake prepared with superfine MP) groups. The sizes and surface properties of superfine MP particles were more homogeneous and smoother than the other samples. As the particle size decreased, the specific volume increased, whereas baking loss, hardness, and chewiness of the sponge cake decreased. Superfine MP and superfine MPS had the highest lutein content. The flavor of marigold and the overall acceptability of sponge cake with superfine MP were 7.90 ± 0.97 and 7.55 ± 0.76, which represents the highest values among the samples. The results of this study have shown that jet milling can contribute to improvements in texture, lutein content, and sensory qualities for baked products with MP.

## 1. Introduction

Age-related macular degeneration (AMD) is a disease affecting individuals of all ages because of increased exposure to blue light [1,2]. Approximately 200 million people experience AMD, which is expected to double by 2040 [3]. Because of this phenomenon, studies have focused on improving eye health; furthermore, increasing lutein intake is a recommended solution to this public eye health problem [4].

Lutein is a yellow pigment that exists in the macula of humans because of carotenoid accumulation in the fovea, a condition called macula lutea [1]. It can improve eye health owing to its chemical structure, which has nine conjugated double bonds in the polyene chain and high absorptivity in terms of blue light; thus, it can act as an effective blue light filter for the retina [1,2]. Unfortunately, unlike animals, humans cannot synthesize lutein, and it must thus be obtained from food [5]. Furthermore, lutein is susceptible to the effects of light, oxygen, and heat; the mono- or di-esterified forms of fatty acids present in lutein and its low water solubility and fat-soluble properties result in its low bioavailability in functional foods [6].

Most studies have focused on fortified lutein content and particle size reduction in raw materials to increase the bioavailability of lutein in the human digestive system with the aim of improving eye health [7]. Particle size reduction influences the water solubility, dissolution rate, and bioavailability of lutein because of an increase in surface area [7]. Among particle size reduction technologies, jet milling, a mechanical particle size reduction method, is commonly used to produce superfine powders; it is not only used in the food industry but also in the pharmaceutical industry [8].

The marigold flower (*Targetes erecta*), which is a primary source of lutein, contains approximately 3.7–5.7 times more lutein than green leafy vegetables; it is also traditionally used as a colorant, dietary supplement, beverage additive, and functional food [9]. The antioxidant activity of lutein has previously been investigated in vitro [10], and encapsulation has been assessed as a method for improving its bioavailability [11]. An increased amount of lutein in marigold flowers is desirable, but the use of lutein-enriched marigold flowers in processed food is lacking. The development of lutein-enriched marigold powder that can increase the lutein content in food products has been considered as an important approach to meeting the demand for eye health-promoting products.

Sponge cake is a bakery product that is consumed globally and contains large amounts of oil [12]. Oil acts as an effective lutein carrier and can promote the micellization of carotenoids while improving the bioaccessibility of lutein [12]. Lutein-enriched marigold flowers can be used to produce baked products with high lutein content and thus meet the nutritional needs related to improving eye health. However, the use of marigold flowers as a source of lutein in food additives has not been studied sufficiently.

Therefore, this study aimed to (1) investigate the properties of lutein-enriched marigold powder through particle size reduction, (2) optimize the quality and sensory properties of baked products with lutein-enriched MP, and (3) assess the possibility of using lutein-enriched MP as a food additive to improve the overall quality and functionality of baked products.

## 2. Materials and Methods

### 2.1. Materials

Lutein-enriched marigold flowers were obtained from WOOREE Green Science Co., Ltd. (Ansan, Republic of Korea) and grown using LED lights to fortify the lutein content. Marvel orange (*Targetes erecta*) was selected for the production of lutein-enriched marigold flowers and grown under artificial conditions (light source: R:G:B = 2:1:1; luminous intensity: 300 μmol/m^2^/s; photoperiod: 11 h; nutrient solution: EC 2.0 dS/m; pH 6.5) for 13 weeks. Fresh flower weight was 6.54–11.84 g/plant depending on harvest time and lutein content (dry basis) was 7.87–11.07 mg/plant. Other ingredients, including wheat flour (carbohydrate 78%, protein 8%), sugar, butter, and salt, were purchased from a local market. All chemicals, solvents, and standards used for chemical analyses were of analytical grade and purchased from Samchun Chemical Co., Ltd. (Gangnam-gu, Korea).

### 2.2. Sample Preparation

Marigold flowers were dried using a forced convection drying oven (SFC-203, Shinsaeng, Paju, Republic of Korea) at 40 °C for 48 h, and only the petals were collected to obtain lutein-enriched marigold flower powder (MP). After being dried, the collected petals were ground and milled to divide them based on their particle size, which was achieved using a sieve shaker (EML200, Haver & Boecker, Oelde, Germany). Two types of sieves were used (150 and 63 µm testing sieves, Nonaka Rikaki, Tokyo, Japan). The lutein-enriched MP was sieved with a 150 µm sieve to collect coarse MP, and the 63 µm sieve was used to obtain fine MP. To obtain superfine MP, the powder was milled using a jet mill (CGS 10, NETZSCH, Selb, Germany).

### 2.3. Determination of Particle Properties

Particle size distribution of the MP samples was determined via the dry method using a laser diffraction particle size analyzer (Mastersizer 3000, Malvern Instrument Limited, Malvern, UK) with a particle size absorption index of 0.1, particle refractive index of 1.53, and dispersant refractive index of 1.0. A scanning electron microscope (S-3400N, Hitachi, Tokyo, Japan) was used to observe the microstructures of the samples. Before measurement, the samples were coated with gold using an ion coater (E-1010; Eiko Co., Kobe, Hyogo, Japan). The color of the samples was measured using a color difference meter (CR-400, Minolta Co., Ltd., Osaka, Japan) according to the CIE (Commission Internationale de Photométrie) *L**, *a**, *b** color system. To determine the color difference between samples, Δ*E*(total color difference) was calculated by Equation (1).
(1)$$\sqrt{{\left(L_{sample}-L_{standard}\right)}^{2}{\left(a_{sample}-a_{standard}\right)}^{2}+{\left(b_{sample}-b_{standard}\right)}^{2}}\;$$

The water-holding capacity (WHC), swelling capacity (SC), and water solubility index (WSI) were measured according to the method proposed by Zhang et al. [13] with slight modifications. Oil-holding capacity (OHC) was determined in accordance with Choi et al. [14]. All values were expressed on a dry weight basis as the average of three measurements.

### 2.4. Sponge Cake Preparation

The samples were prepared with different particle sizes of MP and designated as follows: control (sponge cake prepared without MP), coarse MPS (sponge cake prepared with coarse MP), fine MPS (sponge cake prepared with fine MP), and superfine MPS (sponge cake prepared with superfine MP). Preliminary experiments were performed in accordance with Alotaibi et al. [15] to prepare the formula of the sponge cake. Wheat flour (control: 120 g; MP addition group: 117 g) or 3 g of MP instead of wheat flour was filtered using a 300 µm sieve and added to the MP-enriched group. Eggs (240 g) and sugar (150 g) were then added and mixed at a high speed for 10 min. The prepared wheat flour and egg batter was mixed and then baked in a preheated oven (160 °C, 180 °C) for 25 min.

### 2.5. Determination of the Cooking Properties of Sponge Cake

The moisture content of the samples was measured in accordance with the AACC (2000) method [16] with slight modifications. Briefly, 10 g of the sample was dried in a drying oven at 105 °C for 24 h. Moisture content was calculated as water content percentage per 10 g of the sample. The pH of the supernatant was measured using a pH meter (Seven Easy S20, Mettler Toledo, Greifensee, Switzerland).

The weight of the batter was determined before baking to measure the baking loss (BL) in each sample. After being baked, the samples were cooled at 25 °C for 1 h and weighed.

Specific volume is defined as the number of cubic meters occupied by 1 kg of matter. In this study, the specific volume (L/kg) of the sample was calculated by dividing the volume of sponge cake (mL) by its weight (g). The color of the sample was determined using a color difference meter (Colorimeter, CR-400, Minolta Co., Ltd., Osaka, Japan) according to the CIE *L**, *a**, *b** color system.

### 2.6. Quantification and Extraction of Lutein Content

Lutein was extracted from the samples as described by Liu and Suresh [17] with slight modifications. Briefly, 1 g of the sample was placed in a 50 mL centrifuge tube, mixed with acetone (30 mL), homogenized, and incubated in a water bath at 40 °C for 30 min. Butylated-hydroxy-toluene (10 mg) was added to prevent the oxidation of carotenoids. Afterward, the sample was centrifuged at 8000 rpm for 10 min, and the supernatant was collected. The residue was re-extracted with 30 mL of acetone for 4 h. The acetone extract was collected after centrifugation using the same method. This process was repeated thrice, and the collected acetone layers were transferred into volumetric flasks.

The MP was saponified as described by Vechpanich et al. [18] with slight modifications. The extract solution was allowed to evaporate in a rotary vacuum evaporator at 40 °C for 15 min, and oleoresin was dried in a forced convection drying oven (SFC-203, Shinsaeng, Paju, Republic of Korea) at 40 °C for 12 h. For saponification, 2 mL of ethanol and 0.5 mL of 45% KOH were added per gram of oleoresin. The mixture was vortexed for 1 min and incubated in a water bath at 50 °C for 4 h. The pH of the mixture was adjusted to approximately 7.0 by adding aqueous hydrochloric acid. After saponification, oleoresin was dissolved in 50 mL of ethanol and dispersed by sonication for 15 min. The mixture was then centrifuged at 8000 rpm for 10 min and the supernatant was collected. The residue was treated with 50 mL of ethanol to extract the remaining lutein in the residue. The obtained supernatant was transferred to a volumetric flask and allowed to evaporate in a rotary vacuum evaporator at 40 °C for 15 min. The product was immediately re-dissolved in mobile phase A, passed through a filter (PTFE, 13 mm, Whatman International Ltd., Maidstone, UK), and analyzed using a high-performance liquid chromatography system (HPLC; LC-4000 series, Jasco, Tokyo, Japan) equipped with a UV detector set at 450 nm and a C30 YMC column (250 × 4.6 mm id., 5 mm). Lutein content was determined as described by Derrien et al. [19] with slight modifications.

### 2.7. Sensory Evaluation

The sensory properties of the samples with different MP particle sizes were evaluated through quantitative descriptive analysis. Sensory evaluation panelists were selected from amongst the students of Food Science and Technology at Chung-Ang University. Several tests were conducted to train the panelists. Basic taste solutions at four concentrations were presented to the panelists, who were asked to rate the samples from weak to strong (1%, 3%, 5%, and 10% sucrose solution; 0.02%, 0.05%, 0.1%, and 0.2% citric acid solution; 0.03%, 0.06%, 0.15%, and 0.25% caffeine solution). Thirty students who passed the acuity test were selected as panelists. Specifically, the following sensory properties were evaluated by the panelists using a 9-point scale method: roughness, hardness, sweetness, marigold flavor, and overall acceptability (Table 1).

### 2.8. Statistical Analysis

Data were analyzed through ANOVA in IBM SPSS Statistics 26 (IBM Corp., Armonk, NY, USA) and expressed as mean ± standard deviation. Differences in the mean were determined using Duncan’s multi-range test and considered statistically significant at *p* < 0.05.

## 3. Results

### 3.1. Particle Properties of Differently Sized MP

The particle size distribution of MP samples is shown in Table 2. Dv_10_, Dv_50_, and Dv_90_, which indicate the equivalent diameters at cumulative volumes of 10%, 50%, and 90%, respectively, gradually decreased as particle size decreased (*p* < 0.05). Coarse MP, fine MP, and superfine MP had Dv_50_ values of 315.33 ± 5.77, 119.33 ± 0.58, and 10.40 ± 0.26 μm, respectively. D_[4,3]_ decreased gradually as particle size decreased, indicating that the mean of the volume-weighted diameter decreased [20,21,22]. The use of jet milling to produce defatted soybean flour results in a decrease in Dv_10_, Dv_50_, and Dv_90_ due to a reduction in particle size [20]. The decrease in D_[4,3]_ indicates that the morphological characteristics of the particles remarkably change so that the inner structure of materials can be modified [20,21].

The specific surface area, which indicates the number and diameter of the materials, markedly increased as the particle size decreased. This increase improves the absorption ability of several molecules, such as water; thus, the application of jet milling can effectively and uniformly produce superfine particles (<10 μm) [20].

As shown in Figure 1, the size distribution of superfine MP was more homogeneous than that of coarse and fine MP. Similarly, the particle distribution of vitamin D2 amplifies the white button mushroom powder produced by jet milling; as particle size decreases, Dv_50_ decreases significantly, whereas the specific surface area increases [22]. Reducing particle size can influence the physicochemical properties and bioactive compound extraction rate of grains, vegetables, and flowers because of structural modifications and cell disintegration [21]. In the present study, superfine MP had a more homogeneous particle size distribution and higher specific surface area than the other samples.

### 3.2. Hydration Properties and OHC of Differently Sized MP

The hydration properties (WHC, WSI, and SC) and OHC of differently sized MP are shown in Table 3; the results were generally consistent, with slight differences. Coarse, fine, and superfine MP had a WHC of 10.35 ± 0.55 g/g, 10.22 ± 0.70 g/g, and 5.47 ± 0.44 g/g, respectively. Interestingly, while coarse and fine MP had no significant differences in terms of WHC, superfine MP had a significantly smaller value than the other samples (*p* < 0.05). Caprez et al. [23] demonstrated that large particles are highly hydrated because materials do not form under external force; thus, the complexity of the microstructure can remain intact. Nagashree and Kulkarni [24] also showed that small particles are produced by applying kneading forces and hygroscopic-like forces that weaken intact cellular structures; thus, water easily comes out of materials.

The WSIs of coarse, fine, and superfine MP were 34.67 ± 1.15%, 38.67 ± 1.15%, and 41.33 ± 1.15%, respectively. Consistent with the WSI results, SC significantly increased as particle size decreased. The differences in SC among the materials were likely the result of the properties of their individual components and their physical structures, such as the porosity and crystallinity of their fiber matrix [25].

The hydration properties of a powder determine the texture and functional properties of processed foods and result in different cooking properties [26]. The increase in WSI and SC is probably due to the increase in the surface area of particles and promotes water absorption ability; thus, solubility increases and a soft mouthfeel is achieved [27]. However, the WHC and SC of white button mushrooms decrease when particle size is reduced, whereas the WSI increases. This difference in results could be due to the different properties of food components [22].

Coarse, fine, and superfine MP had an OHC of 3.41 ± 0.03 g/g, 2.96 ± 0.04 g/g, and 2.58 ± 0.06 g/g, respectively. As expected, the particle size and OHC of the samples decreased, possibly because of the damage to the fiber matrix and the collapse of the pores during size-reducing processes such as grinding.

### 3.3. Color and Microstructure of Differently Sized MP

The color values are shown in Table 2. *L** (brightness), *a** (redness), and *b** (yellowness) increased as MP particle size decreased. Coarse, fine, and superfine MP had *L** values of 53.37 ± 0.21, 55.20 ± 0.04, and 66.14 ± 0.02, respectively. Among the samples, superfine MP had the highest *b** value of 57.48 ± 0.18. The value of Δ*E*(total color difference) was the highest in superfine MP, which means that the color difference increased compared to coarse MP. The brightness of materials gradually increases as their particle size decreases because of the increased reflectivity, and the color of materials can be influenced by cell structure, shape, and size [28]. These results are consistent with the observations reported by Muttakin et al. [20], who demonstrated that a decrease in particle size leads to an increase in the brightness of soybean flour. The microstructures of the samples are shown in Figure 2. As the size of the particles decreased, their size, shape, and distribution showed significantly different structural properties. Coarse and fine MP were more structurally intact and had rougher particle surfaces than those of superfine MP. Conversely, superfine MP had a certain size without a specific structure compared to the other samples. These results correlate with the results for specific surface area as the value was significantly increased in superfine MP. These results are also similar to those obtained by Nykamp et al. [29], who used jet milling to produce extremely small particles with smooth surfaces so that they could be used to manufacture drug materials. These results showed that jet milling was effective in breaking down large particles into superfine particles through collisions between particles.

### 3.4. Quality Indicators and Color of the Sponge Cake

The values for moisture content, pH, baking loss (BL), specific volume, and color are listed in Table 3. The moisture content of the samples showed no significant differences. The pH of the samples decreased gradually as MP particle size decreased because of the pH of the MP. The BL of the samples decreased slightly as the size of the particles decreased in the MP-added group. In correlation with the results for BL, the specific volume of the sample increased as the size of particles decreased. As the higher specific volume is caused by an increase in swelling capacity that influences the volume of the sponge cake, the superfine MPS thus showed the highest specific volume among the samples. Large particles have a low density in batter because of coalescence, which induces greater air incorporation than that induced by small particles, thus inducing the low stability of the batter. In addition, the larger the diameter of a particle, the greater the increase in the number of pores, resulting in cakes with a rough texture [30]. As shown in Figure 3, superfine MPS had better pore distribution and smaller pore size than the other samples. A smaller foam size improved the texture, volume, and shape of the baked product. The amount, volume, and distribution of air are major quality parameters in baked products [30,31]. This difference between the samples in this study could be related to the differences in the air incorporation rate, air content, and the pore size of the batter and cake. Auffret et al. [25] reported similar results regarding the addition of soy flour to gluten-free cakes. Specifically, particle size reduction in soy flour induces a decrease in batter density, bubble size, pore size, and the height of cakes. BL increases during baking in correlation with the batter density and height of products. The color, shape, size, and texture of cakes are important factors for evaluating the appearance of baked products. These appearance qualities are incorporated into overall customer experience and satisfaction [32].

The value of *L** significantly decreased as the particle size of MP decreased (*p* < 0.05). The difference between the *L** values of the MP-added group was not significant, but it gradually decreased as particle size decreased; thus, the superfine MP showed the lowest *L** value (53.23 ± 0.66). Among the samples, including the control, superfine MPS (37.82) had the highest Δ*E*. The smaller the particle size, the larger the surface area, which results in increased color intensity. Thus, the color of the sponge cake changed because of the different particle sizes of the MP. Similar results have been reported for the addition of MP to baked pan bread. *L** decreases, whereas *a* and *b* increase; consequently, the produced crust and crumb are darker than the control [15]. In the present study, the color of the sponge cake decreased because of the higher levels of carotenoid pigments in the MP.

### 3.5. Texture Analysis of the Sponge Cake

The textural properties of the samples were evaluated by measuring their hardness, cohesiveness, springiness, gumminess, and chewiness (Table 4). The control, coarse MPS, fine MPS, and superfine MPS had hardness values of 9.31 ± 1.06 N, 11.28 ± 1.26 N, 10.75 ± 0.55 N, and 9.01 ± 0.72 N, respectively. Similar to hardness, the highest chewiness and gumminess values were observed in coarse MPS (6.75 ± 0.72 N and 7.71 ± 0.76 N, respectively); the values of chewiness and gumminess for superfine MPS were the most similar to those of the control. The control and superfine MPS had chewiness of 5.76 ± 0.61 and 5.59 ± 0.53 and gumminess of 6.55 ± 0.74 N and 6.43 ± 0.49 N, respectively. Hardness, chewiness, and gumminess decreased as the particle size of MP decreased, whereas springiness, cohesiveness, and resilience slightly differed without having any clear correlation. In the current study, the improvement in specific surface area and swelling capacity in superfine MP led to a homogeneous distribution of substitution powder which induced softer texture in the sponge cake. This result was due to the substitution of wheat flour with MP, which reduced the gluten content in the sponge cake batter and affected the properties of the sponge cake. These differences were consistent with those reported by [33], who found that the addition of black garlic powder instead of wheat flour increased hardness. Furthermore, the addition of mulberry leaf powder [34] produced similar results, i.e., the substitution of wheat flour increased the hardness and chewiness of bread. Wheat flour maintains the compact inner structures of cakes, which have an intact gluten matrix; thus, the substitution of wheat flour results in a non-uniform pore distribution in cakes, which can lead to an increase in hardness, chewiness, and gumminess [33]. Although hardness, chewiness, and gumminess increased, the addition of superfine MP resulted in the formation of structures similar to those in the control; specifically, superfine MPS had more uniform pore distribution and more compact structures than coarse and fine MPS. This result correlated with the specific volumes of the samples. Some of the differences could be attributed to the WHC, WS, and SC of MP particles. Logically, replacing wheat flour with other materials in baked products causes hardening, which should be avoided. Nevertheless, a reduction in the particle sizes of materials can minimize this negative phenomenon when other materials are used to replace wheat flour.

### 3.6. Determination of Lutein Content

The lutein content of MP is shown in Figure 4. The lutein content of coarse, fine, and superfine MP was 10.99 ± 0.43 mg/g, 13.51 ± 0.34 mg/g, and 14.25 ± 0.44 mg/g, respectively. The lutein content of sponge cake was 9.12 ± 0.10 mg/100, 9.36 ± 0.20 mg/100, and 10.10 ± 0.10 mg/100 g for coarse, fine, and superfine MP, respectively. Therefore, 100 g of cake with MP provided approximately 10 mg of lutein. Alotaibi et al. [15] showed that the lutein content of African and French marigold freeze-dried petals is 150.98 ± 26.87 ppm and 78.26 ± 0.66 ppm, respectively, which is approximately 10 times lower than the lutein content of MP grown under LED lights. This result demonstrated that lutein in MP was relatively stable during baking; thus, significant amounts of lutein were still present in the final baked product. In addition, the reduction in particle size resulted in an increase in the rate of lutein extraction from MP. Consistent with the present study, a previous study demonstrated that lutein content slightly decreases when MP is added to pan bread, suggesting that lutein in MP may be stable during baking [15]. However, the general use of lutein is limited because of its low stability; additionally, it is easily destroyed during food processing, except in bakery products with high oil content, which is a lutein carrier that improves its stability [35]. In the present study, the particle size reduction in MP led to the increased extraction of lutein content in sponge cakes with MP due to the increase in surface area. Furthermore, the increased stability of lutein could be an advantage in high-fat bakery products.

### 3.7. Sensory Evaluation

The results of sensory evaluation are presented in Table 5. The samples had significantly different roughness; specifically, regarding the mouthfeel of the particles, roughness was lowest in the control (1.30 ± 0.73) and highest in coarse MPS (7.85 ± 0.75). Fine and superfine MPS had a roughness of 5.55 ± 0.83 and 2.80 ± 0.77, respectively; thus, the control was similar to superfine MPS but different from coarse and fine MPS. Correlated with roughness, hardness, which indicates the force required to deform a sample, was highest in coarse MPS (7.05 ± 1.10). By comparison, the control, fine MPS, and superfine MPS had a hardness of 1.80 ± 1.20, 4.85 ± 0.88, and 2.45 ± 0.69, respectively. The particle solubility, pore size, and pore distribution of the cake affect its roughness and hardness. Sweetness decreased gradually as particle size decreased. A decline in sweetness can result in a bitter MP taste because of particle size reduction, and more of the phenolic compounds responsible for the bitter taste may be extracted from small particles than from large particles [36]. In contrast, the marigold flavor increased as particle size increased. Although the addition of MP led to a slight decline in the sweetness score, the control and superfine MPS had overall acceptability of 7.20 ± 1.44 and 7.55 ± 0.76, respectively, which did not significantly differ. In the current study, these results indicate that the sweetness and slight differences in roughness and hardness were negatively correlated with the overall acceptability of superfine MPS.

## 4. Conclusions

This study revealed that size reduction of the particles produced by jet milling improved the hydration properties of lutein extracted from MP. When the surface area increased, solubility and swelling in water increased during baking, thereby reducing BL. In correlation with the particle properties of MP, the application of superfine MP in sponge cake produced quality characteristics similar to those of the control, and the addition of superfine MP improved the lutein content of sponge cakes. The breakdown of intact fiber structures induced decreases in WHC, whereas the increase in SC due to the specific surface area reduced baking loss and improved the volume of sponge cake. Furthermore, dramatically increased specific surface area increased the extraction of lutein content; thus, the superfine MP and superfine MPS showed the highest lutein content. However, this phenolic compound imparted a bitter taste to the sponge cake. The addition of health-benefiting compounds, such as phenolic compounds, could result in the development of high-quality functional foods. In terms of lutein content, the added MP would be an attractive characteristic of a functional baked product regardless of particle size. Furthermore, reductions in particle size increased the amount of lutein extracted from the sponge cake and provided stability during baking. Therefore, using superfine MP in sponge cake can effectively increase lutein content without adversely affecting the sensory and textural properties of the products.

## Figures and Tables

**Figure 1 foods-12-00508-f001:**
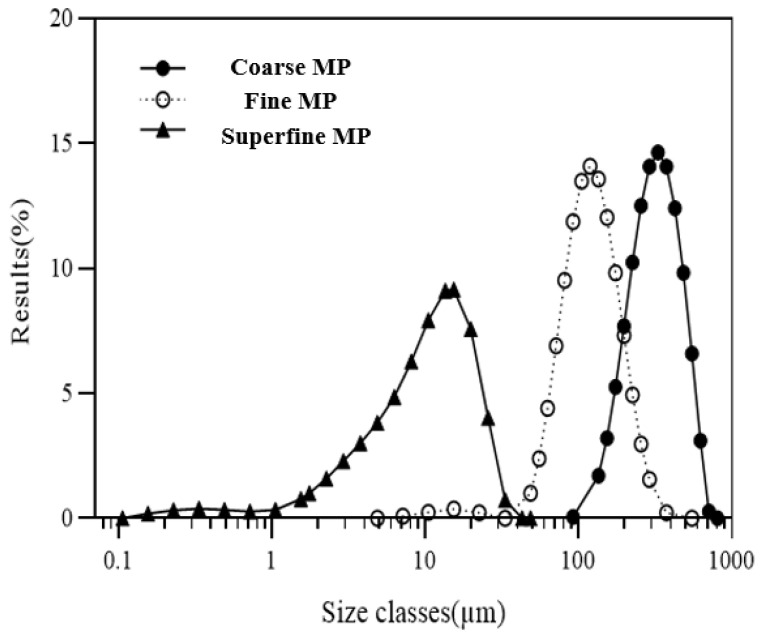
Particle size distribution of differently sized MP. Course MP: marigold course powder. Fine MP: marigold fine powder. Superfine MP: marigold superfine powder.

**Figure 2 foods-12-00508-f002:**
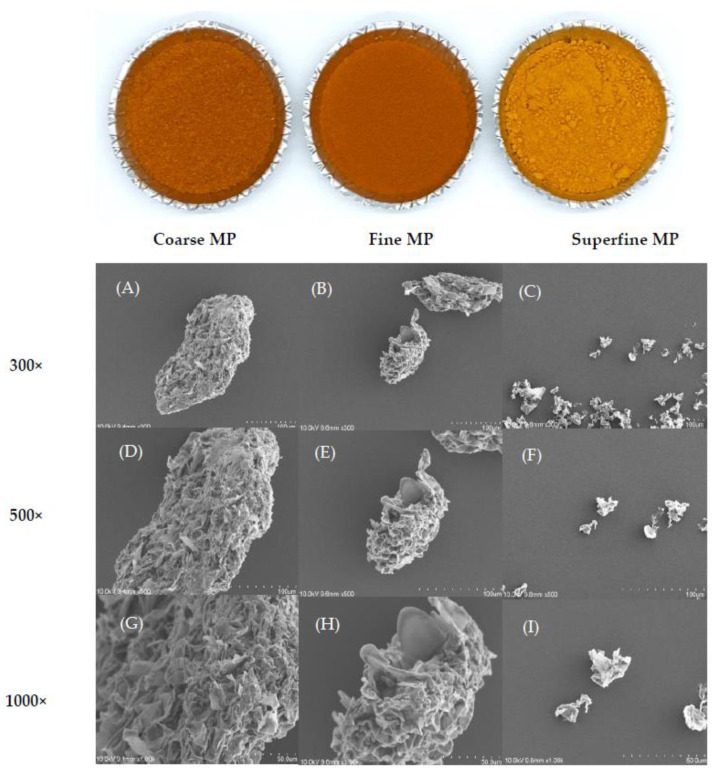
Photographs and SEM images of differently sized MP. Coarse MP: coarse marigold powder. Fine MP: fine marigold powder. Superfine MP: superfine marigold powder. (**A**–**C**) ×300, (**D**–**F**) ×500, (**G**–**I**) ×1000.

**Figure 3 foods-12-00508-f003:**
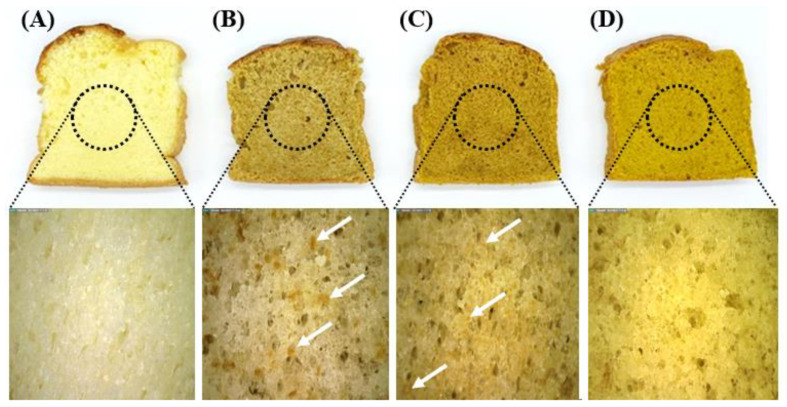
Photographs of sponge cake prepared with differently sized MP. (**A**) Control: sponge cake with MP. (**B**) Course MPS: sponge cake with coarse MP. (**C**) Fine MPS: sponge cake with fine MPS. (**D**) Superfine MPS: sponge cake with superfine MP.

**Figure 4 foods-12-00508-f004:**
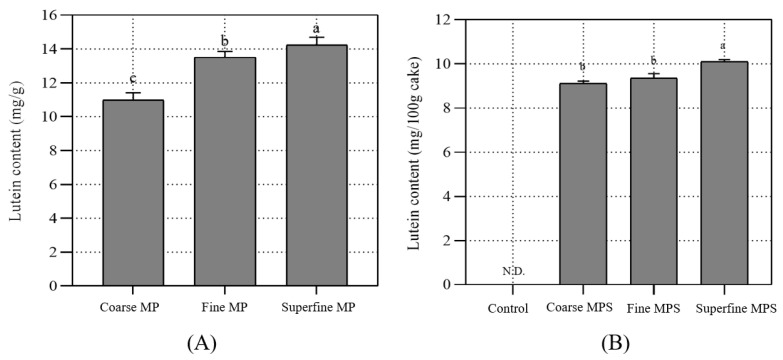
Lutein content of sponge cakes prepared with differently sized MP. (**A**) Lutein content of MP, (**B**) lutein content of sponge cake. Values with different letters within the same column (a–c) are significantly different (*p* < 0.05) according to the results of Duncan’s test. Coarse MP: coarse marigold powder. Fine MP: fine marigold powder. Superfine MP: superfine marigold powder. Control: sponge cake with MP. Coarse MPS: sponge cake with coarse MP. Fine MPS: sponge cake with fine MP. Superfine MPS: sponge cake with superfine MP.

**Table 1 foods-12-00508-t001:** Descriptors used for sensory evaluation of sponge cake with different MP particle sizes through quantitative descriptive analysis.

Descriptor	Definition	References
Roughness	The mouthfeel of the particles	Weak: plain yogurt Strong: corn flour
Hardness	Force required to deform the sample	Weak: cream cheese Strong: chocolate
Sweetness	The fundamental taste sensation produced by the aqueous sucrose solution	Weak: 0 Strong: 10% sucrose solution
Flavor of marigold	Flavor imparted from the added marigold powder in the cake	Weak: 0 Strong: cooked marigold (Flavor)
Overall acceptability	How much the participant liked or did not like the product based on all the sensorial attributes tested	-

**Table 2 foods-12-00508-t002:** Characterization of differently sized MP.

Samples	Particle Size Distribution Properties					Hydration Properties and Oil Holding Capacity				Color			
	Dv_10_ (μm)	Dv_50_ (μm)	Dv_90_ (μm)	D_[4,3]_ (μm)	Specific Surface Area (m^2^/kg)	WHC (g/g)	WSI (%)	SC (mL/g)	OHC (g/g)	*L**	*a**	*b**	Δ*E*
Coarse MP ^(2)^	175.67 ± 0.58 ^a(1)^	315.33 ± 5.77 ^a^	505.33 ± 7.51 ^a^	326.33 ± 4.93 ^a^	95.67 ± 1.26 ^b^	10.35 ± 0.55 ^a^	34.67 ± 1.15 ^c^	11.47 ± 0.76 ^b^	3.41 ± 0.03 ^a^	53.37 ± 0.21 ^c^	25.23 ± 0.41 ^c^	32.96 ± 0.61 ^c^	-
Fine MP	68.60 ± 0.17 ^b^	119.33 ± 0.58 ^b^	205.00 ± 0.00 ^b^	128.67 ± 0.58 ^b^	210.43 ± 1.31 ^b^	10.22 ± 0.70 ^a^	38.67 ± 1.15 ^b^	12.53 ± 0.12 ^b^	2.96 ± 0.04 ^b^	55.20 ± 0.04 ^b^	26.39 ± 0.02 ^b^	35.34 ± 0.24 ^b^	3.22
Superfine MP	2.59 ± 0.08 ^c^	10.40 ± 0.26 ^c^	22.17 ± 0.74 ^c^	11.53 ± 0.32 ^c^	5084.00 ± 168.88 ^a^	5.47 ± 0.44 ^b^	41.33 ± 1.15 ^a^	14.20 ± 1.00 ^a^	2.58 ± 0.06 ^c^	66.14 ± 0.02 ^a^	27.59 ± 0.04 ^a^	57.48 ± 0.18 ^a^	27.75

^(1)^ Values with different letters within the same column (a–c) are significantly different (*p* < 0.05) according to the results of Duncan’s test. ^(2)^ Coarse MP: coarse marigold powder. Fine MP: fine marigold powder. Superfine MP: superfine marigold powder.

**Table 3 foods-12-00508-t003:** Physical properties of sponge cake prepared with differently sized MP.

Samples ^(3)^	Moisture Content (%)	pH	Baking Loss (%)	Specific Volume (L/kg)	Color			
					*L**	*a**	*b**	Δ*E*
Control	36.77 ± 0.68 ^NS(1)^	8.10 ± 0.05 ^a(2)^	5.15 ± 0.53 ^b^	2.88 ± 0.20 ^a^	83.36 ± 0.41 ^a^	0.21 ± 0.05 ^d^	31.07 ± 0.27 ^c^	
Coarse MPS ^(2)^	37.09 ± 0.62	7.81 ± 0.02 ^b^	6.14 ± 0.49 ^a^	2.50 ± 0.09 ^b^	57.83 ± 0.85 ^b^	6.15 ± 0.30 ^c^	30.48 ± 0.80 ^c^	26.32
Fine MPS	37.69 ± 0.27	7.70 ± 0.03 ^c^	5.56 ± 0.60 ^ab^	2.68 ± 0.04 ^ab^	54.04 ± 0.85 ^c^	8.65 ± 0.37 ^b^	36.12 ± 0.73 ^b^	31.04
Superfine MPS	36.96 ± 0.12	7.65 ± 0.02 ^d^	4.94 ± 0.19 ^b^	2.89 ± 0.20 ^a^	53.23 ± 0.66 ^d^	9.90 ± 0.22 ^a^	51.58 ± 2.18 ^a^	37.82

^(1)^ NS: no significant difference; ^(2)^ values with different letters within the same column (a–d) are significantly different (*p* < 0.05) according to the results of Duncan’s test; ^(3)^ control: sponge cake with MP; coarse MPS: sponge cake with course MP; fine MPS: sponge cake with fine MPS; superfine MPS: sponge cake with superfine MP.

**Table 4 foods-12-00508-t004:** The quality characteristics of sponge cake prepared with differently sized MP.

Samples	Hardness (N)	Springiness	Cohesiveness	Chewiness	Gumminess (N)
Control ^(3)^	9.31 ± 1.06 ^b(2)^	0.88 ± 0.01 ^NS(1)^	0.63 ± 0.00 ^a^	5.76 ± 0.61 ^NS^	6.55 ± 0.74 ^ab^
Coarse MPS	11.28 ± 1.26 ^a^	0.87 ± 0.01	0.60 ± 0.01 ^b^	6.75 ± 0.72	7.71 ± 0.76 ^a^
Fine MPS	10.75 ± 0.55 ^ab^	0.90 ± 0.01	0.63 ± 0.00 ^a^	6.72 ± 0.44	7.53 ± 0.38 ^ab^
Superfine MPS	9.01 ± 0.72 ^b^	0.87 ± 0.02	0.64 ± 0.00 ^a^	5.59 ± 0.53	6.43 ± 0.49 ^b^

^(1)^ NS: no significant difference; ^(2)^ values with a different letter within the same column (a,b) are significantly different (*p* < 0.05) according to the results of Duncan’s test. ^(3)^ Control: sponge cake with MP; coarse MPS: sponge cake with coarse MP; fine MPS: sponge cake with fine MPS; superfine MPS: sponge cake with superfine MP.

**Table 5 foods-12-00508-t005:** Sensory evaluation of sponge cake prepared with differently sized MP.

Samples	Roughness	Hardness	Sweetness	Flavor of Marigold	Overall Acceptability
Control ^(2)^	1.30 ± 0.73 ^d(1)^	3.65 ± 0.99 ^c^	7.55 ± 1.05 ^a^	1.00 ± 0.00 ^d^	7.20 ± 1.44 ^a^
Coarse MPS	7.85 ± 0.75 ^a^	5.70 ± 0.66 ^a^	6.30 ± 0.86 ^b^	3.90 ± 1.02 ^c^	3.90 ± 1.17 ^c^
Fine MPS	5.55 ± 0.83 ^b^	4.60 ± 0.88 ^b^	5.25 ± 0.72 ^c^	5.40 ± 0.94 ^b^	5.65 ± 1.18 ^b^
Superfine MPS	2.80 ± 0.77 ^c^	3.80 ± 0.70 ^c^	3.75 ± 0.85 ^d^	7.90 ± 0.97 ^a^	7.55 ± 0.76 ^a^

^(1)^ Values with different letters within the same column (a–d) are significantly different (*p* < 0.05) according to the results of Duncan’s test; ^(2)^ control: sponge cake with MP; coarse MPS: sponge cake with course MP; fine MPS: sponge cake with fine MPS; superfine MPS: sponge cake with superfine MP.

## Data Availability

The data presented in this study are available on request from the corresponding author.

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
