# Peer review of "Superfine Marigold Powder Improves the Quality of Sponge Cake: Lutein Fortification, Texture, and Sensory Properties"

_foods, 2023, doi:10.3390/foods12030508_

Round 1

Reviewer 1 Report

Comments for Authors

I reviewed the current manuscript entitled “Superfine marigold powder improves the quality of sponge cake: lutein fortification, texture, and sensory properties”. The current paper investigates the properties of lutein-enriched marigold powder with particle size reduction, optimizes the quality and sensory properties of baked products with lutein-enriched marigold flower powder. The paper is interesting in its subjects and assesses the possibility of using lutein-enriched marigold flower powder as a food additive to improve the overall quality and functionality of baked products. However, some remarks for further improvement are;

Abstract

Line 19-20, significantly??? Decrease or increase

Line 21-22, rephrase for clarity. For sensory evaluation, the score should be mentioned instead of the increasing trend.

Choose appropriate keywords for the attention of researchers/readers. Add more keywords that are not contained in the title or the abstract.

Introduction

This section could be improved. Some data must be added so more focus can be given to the Lutein sources, composition, nutritional studies, etc. Please mention clinical studies regarding the health-beneficial effects of lutein for human beings.

Materials and Methods

The author must have studied the Marigold flower powder chemical composition, with that author could calculate and predict the results regarding the functional and technological properties of the lutein-enriched marigold powder.

Line 97. CIE? An explanation should be added to the abbreviation first.

To the materials and methods part: line 113 the AACC method number must be mentioned; the preparation part must be better described

Results

The reasoning for each result was inadequate, such as functional properties, microstructure, volume, etc. Coarse, fine, and superfine MP should be studied comparatively.

The optical quality of the SEM images is very low. Additionally, no scale bars are shown.

Line 252, BL stands for?

Line 271, rephrase the statement “Color and L gradually decreased as the particle size of MP decreased.” for better understanding.

Table 2. cited in text “3.3. Color and microstructure of differently sized MP”. Table 2 should be placed after citing.

Conclusion

Conclusions must be more focused and targeted on the overall outcome of the study. The present section is a summary of the results.

The script must be improved in terms of language, grammar, and syntax. 

Author Response

Journal

Foods

Title

Superfine marigold powder improves the quality of sponge cake: lutein fortification, texture, and sensory properties

Dear Editor,

The authors are grateful to the editor and reviewers for their interest in the findings and valuable suggestions that helped improve the manuscript. A point-by-point response to the editor’s comments is included below. All changes in the revised manuscript are highlighted in red text font.

No.

question

line

answer

line

1

significantly??? Decrease or increase

19-20

Thank you for your comments. We corrected this part.

19-20

2

rephrase for clarity. For sensory evaluation, the score should be mentioned instead of the increasing trend.

21-22

Thank you for your comments. We corrected this part.

=> The flavor of marigold and overall acceptability of sponge cake with superfine MP were 7.90±0.97 and 7.55±0.76 which is the highest value among the sample.

21-23

3

Choose appropriate keywords for the attention of researchers/readers. Add more keywords that are not contained in the title or the abstract.

25

Thank you for your comments. We corrected this part.

=>Jet mill; Lutein-enriched marigold powder; Baking; texture profiles; Sponge cake; Sensory evaluation

25-26

4

The author must have studied the Marigold flower powder chemical composition, with that author could calculate and predict the results regarding the functional and technological properties of the lutein-enriched marigold powder.

Materials and Methods

Thank you for your comments.

We corrected this part.

=>The fresh flower weight was 6.54-11.84 g/plant dependent on harvest time and the lutein content (dry basis) was 7.87-11.07 mg/plant.

76-78

5

CIE? An explanation should be added to the abbreviation first.

97

We corrected this part.

=> CIE (Commission Internationale de Photométrie) L*, a*, b* color system

99

7

To the materials and methods part: line 113 the AACC method number must be mentioned; the preparation part must be better described

113

Thank you for your comments.

We corrected this part.

119

8

The reasoning for each result was inadequate, such as functional properties, microstructure, volume, etc. Coarse, fine, and superfine MP should be studied comparatively.

Results

Thank you for your comments.

We corrected this part.

248

280

372

9

The optical quality of the SEM images is very low. Additionally, no scale bars are shown.

246

Thank you for your comments.

We corrected this part.

271

10

BL stands for?

252

Thank you for your comments. We corrected this part.

=>The moisture content, pH, Baking Loss, specific volume, and color are listed in Table 3.

250

11

rephrase the statement “Color and L gradually decreased as the particle size of MP decreased.” for better understanding.

271

Thank you for your comments. We corrected this part.

=>The value of L* significantly decreased as the particle size of MP decreased (p<0.05).

12

Table 2. cited in text “3.3. Color and microstructure of differently sized MP”. Table 2 should be placed after citing.

Table 2

Thank you for your comments.

=> But in line 171, Table 2 is required for describing particle distribution properties thus it should be placed in line 196.

196

13

Conclusions must be more focused and targeted on the overall outcome of the study. The present section is a summary of the results.

Thank you for your comments.

We corrected this part.

=> The breakdown of intact fiber structure induces decreased in WHC whereas the increase in SC due to the specific surface area reduced baking loss and improve the volume of sponge cake. Furthermore, dramatically increased in specific surface area increases the extraction of lutein contents thus the superfine MP and superfine MPS showed the highest contents of lutein.

412

14

The script must be improved in terms of language, grammar, and syntax. 

Thank you for your comments.

=> We re-edited by Editage (www.editage.co.kr) for English language editing.

Reviewer 2 Report

Dear Authors,

I rate the manuscript submitted for review as very good. Below I present some minor remarks to the text.

line 50: Marigold flower (Targetes erecta),…

line 73-74: Marvel orange  Marigold (Tagetes erecta)

line 107: Wheat flour - please provide the parameters of the flour used, gluten and ash content.

line 122-123: In this study, the specific volume (L/kg) of the sample was calculated by dividing the volume of bread sponge cake (mL) by its weight (g).

line 125: It would also be useful to calculate the total color difference between the control sample and the test sample (ΔE).

line 166: Data were analyzed through ANOVA in IBM SPSS Statistics 26 (producer, country) and expressed as…

Table 2: The table shows the result of ΔE calculations, here this parameter is redundant, what was of the standard of color (control sample)?

line 249: “3.4. Cooking properties Quality indicators and color of the sponge cake”

line 274-275: What criterion for the interpretation of ΔE value was used in the discussion of the results of study?

Figure 1, 2: This is an interesting and well-prepared presentation of research results.

line 319-320: “Some of the differences could be attributed to the WHC, WS, and SC of MP particles.” The issues related to the listed properties of the additives are equally important here. Sponge cake is not a typical wheat dough (for bread). Sponge cake of the same good quality as wheat can be obtained from gluten-free flours. In creating this type of dough, the idea is to aerate the mass (flour-eggs-sugar) by whipping the dough or using chemical leavening agents. I suggest you pay attention to the possible "competition" in the test sample dough (batter) ... starch, gluten from flour and marigold flower powder (2.5% of flour mass).

Kind regards

Reviewer

Author Response

Journal

Foods

Title

Superfine marigold powder improves the quality of sponge cake: lutein fortification, texture, and sensory properties

Dear Editor,

The authors are grateful to the editor and reviewers for their interest in the findings and valuable suggestions that helped improve the manuscript. A point-by-point response to the editor’s comments is included below. All changes in the revised manuscript are highlighted in red text Marigold flower (Targetes erecta), font.

No.

question

line

answer

line

1

Marigold flower (Targetes erecta),…

50

Thank you for your comments.

We corrected this part.

50

2

Marvel orange  Marigold (Tagetes erecta)

73-74

Thank you for your comments.

We corrected this part.

75

3

Wheat flour - please provide the parameters of the flour used, gluten and ash content.

107

Thank you for your comments.

We corrected this part

=> wheat flour(carbohydrate 78%, protein 8%),

77

4

In this study, the specific volume (L/kg) of the sample was calculated by dividing the volume of bread sponge cake (mL) by its weight (g).

122-123

Thank you for your comments.

We corrected this part

=> In this study, the specific volume of the sample was calculated by dividing the volume of sponge cake (mL) by its weight (g).

124

5

 It would also be useful to calculate the total color difference between the control sample and the test sample (ΔE).

125

Thank you for your comments.

We corrected this part

=>The color of the samples was measured using a color difference meter (CR-400, Minolta Co., Ltd., Japan) according to the CIE L*, a*, b* color system. To determine the color difference between the sample, ΔE(total color difference) was calculated by Eq. (1).

99-100

7

Data were analyzed through ANOVA in IBM SPSS Statistics 26 (producer, country) and expressed as…

166

Thank you for your comments.

We corrected this part

=> Data were analyzed through ANOVA in IBM SPSS Statistics 26 (IBM Corp., Armonk, NY) and expressed as mean ± standard deviation.

170

8

Table 2: The table shows the result of ΔE calculations, here this parameter is redundant, what was of the standard of color (control sample)?

Table2

Thank you for your comments.

=>The standard of color is a coarse sample.

Table2

9

“3.4. Cooking properties Quality indicators and color of the sponge cake”

249

Thank you for your comments.

We corrected this part.

249

10

What criterion for the interpretation of ΔE value was used in the discussion of the results of study?

274-275

The value of ΔE shows the color difference compared to the coarse sample. The higher the value, the larger the color difference obtained. Therefore, it is the general parameter to compare the sample color difference between the sample. We corrected this part.

-> The value of ΔE(total color difference) was the highest in superfine MP which means that the color difference increased compared to Coarse MP.

236

11

Figure 1, 2: This is an interesting and well-prepared presentation of research results.

271

Thank you for your comments.

12

 “Some of the differences could be attributed to the WHC, WS, and SC of MP particles.” The issues related to the listed properties of the additives are equally important here. Sponge cake is not a typical wheat dough (for bread). Sponge cake of the same good quality as wheat can be obtained from gluten-free flours. In creating this type of dough, the idea is to aerate the mass (flour-eggs-sugar) by whipping the dough or using chemical leavening agents. I suggest you pay attention to the possible "competition" in the test sample dough (batter) ... starch, gluten from flour and marigold flower powder (2.5% of flour mass).

319-320

Thank you for your comments.

We corrected this part.

We compare each value in the test sample but without a mention of starch, or gluten from flour because we do not investigate the starch or gluten content of marigold flour and wheat flour. Although we don’t know about gluten or starch contents, we reviewed about the texture difference with the result of the current study. Please check about this part. Thank you.

336
